# Ephedrine and pseudoephedrine in *Ephedra saxatilis* on the vertical altitude gradient changed in southern Tibet Plateau, China

**Mengnan Lu‡, Yongjuan Zhang‡, Shiyan Wang, Xiaona Wang, Shengnan Zhang, Ji De** ⓘ *

School of Ecology and Environment, Tibet University, Lhasa, Tibet, China

‡ ML and YZ share first authorship on this work.
* dekyi1981@utibet.edu.cn

**Data Availability Statement:** All relevant data are within the paper and its Supporting Information files.

## Abstract

*Ephedra* is one of the world's most important plants, used in medicine, plants and ecology. Most *Ephedra* grows in plain areas and is stable. But the plateau environment is special, with the change of altitude, the variety difference of plateau *Ephedra saxatilis* is very obvious. *E. saxatilis* metabolism on the Tibetan Plateau is not only affected by altitude, but also environmental conditions such as climate conditions and different soil components. However, the change mechanism of *E. saxatilis* alkaloids in special ecological environment is still unclear. Therefore, we analyzed the metabolic and altitude of *E. saxatilis* species in the Tibetan Plateau. Through the functional analysis of Kyoto Metabolism and Metabolomic Encyclopedia (KEGG), we can determine that the number of *E. saxatilis* metabolites decreases with the increase of altitude, and there are differences in metabolism among the three mountains. This was confirmed by univariate analysis of the top five metabolic pathways. Based on the analysis of soil and metabolomics, it was found that soil water content was also a factor affecting *E. saxatilis* metabolism. According to the difference of vertical height gradient, ephedrine and pseudephedrine showed the same change in vertical altitude under different mountains. Ephedrine increased as the altitude gradient increased, and pseudoephedrine decreased as the altitude gradient decreased. Our results provide valuable information for further study of metabolic mechanism and efficacy stability. It provides useful reference for the research of *E. saxatilis* planting in special area.

## 1 Introduction

The medicinal resources of *Ephedra* are very rich, the metabolites in the body such as ephedrine, polysaccharide, flavonoids and so on are very rich [1]. The stems of some *Ephedra* (ephedraceae) are rich in metabolites such as ephedrine and pseudoephedrine, effective for headaches, bronchial asthma, rhinitis, and the common cold [2]. With the research of *Ephedra* more and more in-depth, people found that *Ephedra* contains a variety of chemical components, such as volatile oil, flavonoids, polysaccharides, amino acids, tannins, organic acids, lignans and trace elements, but alkaloids is still the most important medicinal ingredients of *Ephedra* [3–5].

**Funding:** Our work was partly financially supported by the National Natural Science Foundation of China (32060087) and Tibet University supporting and cultivating project (2022:1). J.D., M.-N.L., Y.-J. Z., S.-N.Z., S.-Y.W. and X.-N.W participated in this fund. The funders had role in study design, data collection and analysis, decision to publish, or preparation of the manuscript.There was no additional external funding received for this study.

**Competing interests:** The authors have declared that no competing interests exist.

*Ephedra* is distributed in 40 groups all over the world, such as Japan, Nepal, Uzbekistan, China and so on [6–9]. There are about 6 species in the southern Tibet Plateau of China. The Tibetan Plateau and its adjacent areas have strong ecosystem diversity and vegetation is obviously affected by local climate and topography [10]. Previous studies on *Ephedra* mostly focused on the phylogeographic structure, phylogenetic reconstruction and chemical substances [11, 12]. However, ecological investigation of the effects of altitude on alkaloids has not been established. *E. saxatilis*, which is unique to Tibet in China, *E. saxatilis* is difficult to pick due to its high altitude, so the analysis of ephedrine and pseudoephedrine can find the factors that determine its stable efficacy, which can provide references for our later cultivation of *E. saxatilis* and provide a stable source for medicinal materials in Tibet.

Altitude as a unique geographical environment in Tibet, many plants are at different altitudes in Tibet, so their morphology and internal composition differ [13, 14]. The content of ephedrine and pseudoephedrine in *Ephedra* varies significantly, which is not only related to genetic factors, but also related to habitat conditions [15–17]. *E. saxatilis* grows at different altitudes, and its internal metabolism changes with altitude are not known. Minami [7] find alkaloid determination of *Ephedra* in different habitats revealed the relationship between the chemical constituents of *Ephedra* in different growing environments. *Ephedra* is rich in substances and complex in composition. Therefore, metabolomics is a good method to detect and screen *Ephedra* in large quantities, and to reveal the changes of ephedrine and pseudoephedrine from the metabolic level [18, 19]. With the difference of altitude gradient, the mechanism of *E. saxatilis* metabolic rate affecting ephedrine and pseudoephedrine has not been studied. Therefore, the metabolomics method was adopted in this study to identify the change of ephedrine and pseudoephedrine through the relationship between the metabolic rate of *E. saxatilis* and environmental factors. In this context, the metabolism and ecology of *E. saxatilis* were studied to find out the trend of ephedrine and pseudoephedrine with altitude change, and to provide theoretical basis for sustainable utilization and industrialization of *E. saxatilis* in the Qinghai-Tibet Plateau and surrounding areas.

## 2 Materials and methods

### 2.1 Plant material and environment data collection

In our survey of *E. saxatilis* on the Tibetan Plateau, we selected *E. saxatilis* communities growing under the same climatic conditions in the Xigaze and Lhasa region of the southern Tibetan Plateau in 2021–2022 at vertical altitudes. The *Ephedra* in this sampling area has been identified by doctor La qiong as *E. saxatilis* reference to [10, 15, 20], they are predicted to have different metabolite compositions. We chose a mountain with a vertical elevation gradient within 500 meters to collect *E. saxatilis*, because the test data would be more accurate and reliable in a similar elevation gradient range. As shown in Table 1, the locations and characteristics of the S1-S3 communities in Shangyadong Township, Yadong County, southern Tibet were all located at 4021–4500 m near Zhuomulari Snow Mountain (7326 m above sea level), the highest mountain on the Lassa-Yadong Expressway. Between the Chalmulari and Nathu La Mountains. The locations and characteristics of the S4-S6 communities in Jilong County, mountain of Drepung Monastery in Lhasare located at an altitude of 3671-4152m. The locations and characteristics of the S7-S9 communities in Jilong County, located at an altitude of 3721-3986m, in the Mara Mountains in the north of the county. Based on the classification of vegetation in the Tibetan Plateau, the vegetation of each community habitat was divided [21, 22].

*E. saxatilis* specimens collected from each habitat are shown in Table 1 and Fig 1. Cut the land portion from the entire stock without regard to the sex of the stock or the presence of cones. The age of all *E. saxatilis* plants growing in this study area is unknown, but the

**Table 1. Community location and vegetation characteristics of *E. saxatilis* in southern Tibet Plateau.**

| Specimen | Mountain | Collection dates | Number of samples collected | Longitude/latitude | Altitude/m | Vegetation |
|---|---|---|---|---|---|---|
| S1 | 1 | 1 August 2021 | 9 | 27°62′23″ N / 89°04′17″ E | 4025 | Alpine Screes |
| S2 | | 1 August 2021 | 10 | 27°62′16″ N / 89°04′28″ E | 4210 | Alpine Screes |
| S3 | | 1 August 2021 | 9 | 27°62′37″ N / 89°04′19″ E | 4484 | Alpine Screes |
| S4 | 2 | 5 August 2022 | 12 | 29°67′23″ N / 91°05′99″ E | 3671 | Alpine steppe |
| S5 | | 5 August 2022 | 11 | 29°67′29″ N / 91°05′44″ E | 3943 | Alpine steppe |
| S6 | | 5 August 2022 | 8 | 29°67′53″ N / 91°05′11″ E | 4152 | Alpine steppe |
| S7 | 3 | 11 August 2022 | 10 | 28°43′21″ N / 85°58′28″ E | 3721 | Alpine meadow |
| S8 | | 11 August 2022 | 7 | 28°43′61″ N / 85°58′82″ E | 3855 | Alpine meadow |
| S9 | | 11 August 2022 | 10 | 28°43′04″ N / 85°58′39″ E | 3986 | Alpine meadow |

terrestrial portion of all *E. saxatilis* specimens is the same size about 10 cm tall and about 10 cm in diameter. All plant specimens are stored in the School of Science, Xizang University. Community soil samples were collected from 10 to 20 cm above the surface.

## 2.2 Metabolite profiling and metabolomics

**2.2.1 *Sample preparation* for metabolism.** Analysis of metabolites in *E. saxatilis* by Ren [23] and Wang [24] with some modifications. 50 mg solid sample was added to a 2 mL centrifuge tube and a 6 mm diameter grinding bead was added. 400 μL of extraction solution (methanol: water = 4:1 (v:v)) containing 0.02 mg/mL of internal standard (L-2-chlorophenylalanine) was used for metabolite extraction. Samples were ground by the Wonbio-96c (Shanghai wanbo biotechnology co., LTD) frozen tissue grinder for 6 min (-10°C, 50 Hz), followed by low-temperature ultrasonic extraction for 30 min (5°C, 40 kHz).

The samples were left at -20°C for 30 min, centrifuged for 15 min (4°C, 13000 g), and the supernatant was transferred to the injection vial for LC-MS/MS analysis.

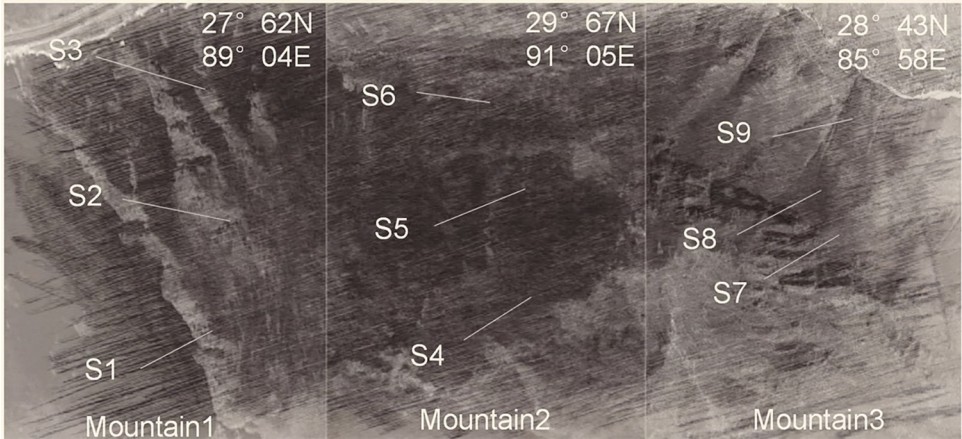

**Fig 1. Coordinates and landform of the collection site.** (Note: There are three sampling points in Mountain 1: S1,S2 and S3, and the altitude range of sampling points is 4025-444m. There are three sampling points S4,S5 and S6 in Mountain 2, with elevations ranging from 3671 to 4152m. There are 3 sampling points S7,S8 and S9 in Mountain 3, and the altitude range is 3721-3986m.)

**2.2.2 MS parameters.** The mass spectrometric data was collected using a Thermo HPLC-Q Exactive Mass Spectrometer equipped with an electrospray ionization (ESI) source operating in either positive or negative ion mode. The optimal conditions were set as followed: heater temperature, 400˚C; Capillary temperature, 320˚C; sheath gas flow rate, 40 arb; Aux gas flow rate, 10 arb; ion-spray voltage floating (ISVF), -2800V in negative mode and 3500V in positive mode, respectively; Normalized collision energy, 20-40-60V rolling for MS/MS. Full MS resolution was 70000, and MS/MS resolution was 17500. Data acquisition was performed with the Data Dependent Acquisition (DDA) mode. The detection was carried out over a mass range of 70–1050 m/z.

## 2.3 Data processing

Limma software in the R statistical package (**http://www.r-project.org**) was used for screening. Differential metabolites among groups were summarized and mapped into their biochemical pathways through metabolic enrichment and pathway analysis based on a database search (KEGG, **http://www.genome.jp/kegg/**). The data were analyzed through the free online platform of the majorbio cloud platform (cloud.majorbio.com).The data were analyzed using two-way analysis of variance (ANOVA) with treatment and time as factors. The means were separated by Duncan's multiple range test, and differences at $P < 0.05$ were considered to be significant.

## 3 Results

### 3.1 Analysis of main composition of soil

The 9 *E. saxatilis* communities are distributed on three mountains as shown in Table 2. The vegetation types on the three mountains were different, with Mountain 1 as alpine meadow, Mountain 2 as alpine steppe, and Mountain 3 as alpine Screes. The soil moisture ranges from 0.14 to 0.17 $mm^3/mm^3$ for Mountain 1, 0.09 to 0.15 $mm^3/mm^3$ for Mountain 2, and 0.07 to 0.11 $mm^3/mm^3$ for Mountain 3. The temperature of Mountain 1 is 23˚C, the temperature of mountain 2 is 22˚C, and the temperature of mountain 3 is 19˚C. The soil color on all three mountains is brownish black. The loss on ignition of Mountain 1 is (10.7±0.5–16.4±0.4)%, the

**Table 2. Soil information of *Ephedra* collection sites in three mountains.**

| Specimen | Mountain 1 | Mountain 2 | Mountain 3 |
|---|---|---|---|
| Vegetation | Alpine Screes | Alpine steppe | Alpine meadow |
| Soil moisture content | 0.07–0.11 $mm^3/mm^3$ | 0.09–0.15 $mm^3/mm^3$ | 0.14–0.17 $mm^3/mm^3$ |
| Temperature/˚C | 23 | 22 | 19 |
| Loss on ignition (%) | 7.6±0.2–11.7±0.6 | 8.2±0.4–15.3±0.2 | 10.7±0.5–16.4±0.4 |
| Soil color | Brownish black | Brownish black | Brownish black |
| Particle-size distribution in soil (%) | | | |
| Clay | 6.9 | 5.7 | 6.2 |
| Silt | 12.9 | 66.4 | 54.2 |
| Sand | 80.2 | 27.9 | 39.6 |
| Soil texture | Slity loam | Slity loam | Sandy loam |

(Location of each collection site by site code is shown in Table 1

Vegetation at each collection site was identifed based on classifications of the vegetation of Tibet Plateau

The ignition loss is analyzed for 3 times, and the mean value and standard deviation are given

Soil texture was defined using the soil texture classification system of the International Soil Science Society)

soil loss of Mountain 2 is (8.2±0.4–15.3±0.2)%, and the soil loss of Mountain 1 and Mountain 3 is (7.6±0.2–11.7±0.6)%. The soil composition of Mountain 1 and Mountain 2 is similar, and the highest content of silt, which is slity loam. Mountain 3 has the highest sand, which is sandy loam.

## 3.2 PLS-DA analysis and inspection of principal component in *Ephedra saxatilis*

In the PLS-DA model, the values of R2Y and Q2 are often used to judge its stability and observe the reliability of the original data. Generally, the values of both are close to 1 in successful models. As shown in Fig 2(A), R2Y = 0.79, Q2 = 0.75, indicating that the model can well explain the differences among samples. In order to prevent overfitting of the model, the model was verified by Response Permutation Testing (RPT test) for 200 times, in which R2 and Q2 were intercept values of the regression line and Y-axis, and R2 represented the sum of variances that could be explained by the model. Q2 represents the predictive ability of the model. When using the RPT test, Q2 is generally required to be less than zero. As can be seen from Fig 2(B), R2 = 0.569, Q2 = -0.257, the regression line shows an upward trend, Q2 is less than zero, there is no overfitting phenomenon, the model is reliable and can be used for subsequent analysis.

## 3.3 Analysis of total metabolites in community samples of *Ephedra saxatilis*

Cluster analysis was conducted on the detected metabolites of 9 communities in the three mountains, as shown in Fig 3(A). In Mountain 1, the detected metabolites of S2 and S3 were similar, but different from those of S1. The metabolite composition of S5 and S6 in Mountain 2 was detected to be similar and different from that of S4, while the metabolite composition of S8 and S9 in Mountain 3 was detected to be similar and different from that of S9. These results indicate that there are significant differences in metabolic levels among the three mountains. As shown in Fig 3(B), 55 of the same metabolites were detected in three samples of Mountain 1, 72 of the same metabolites were detected in three samples of Mountain 2, and 36 of the same metabolites were detected in three samples of Mountain 3. The results showed that the metabolism levels of *E. saxatilis* communities in the three mountains were not consistent. As

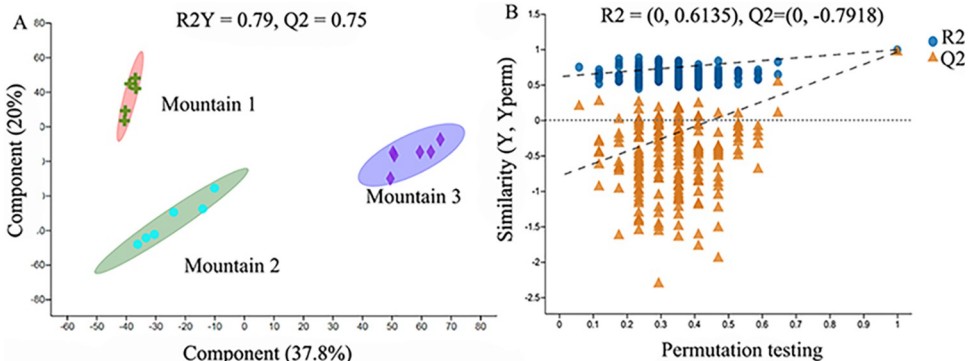

**Fig 2.** Partial least squares analysis (PLS-DA) was used to analysis Ephedra saxatilis samples (A). Permutation test plots of PLS-DA model (B). Taken together, these two axes explain the overall distribution of metabolites in the three communities. Note: The distance of each coordinate point represents the degree of aggregation and dispersion between samples. The closer the distance is, the higher the similarity between samples, and the farther the distance is, the greater the difference between samples. The confidence ellipse indicates that this group of samples is at 95% confidence.

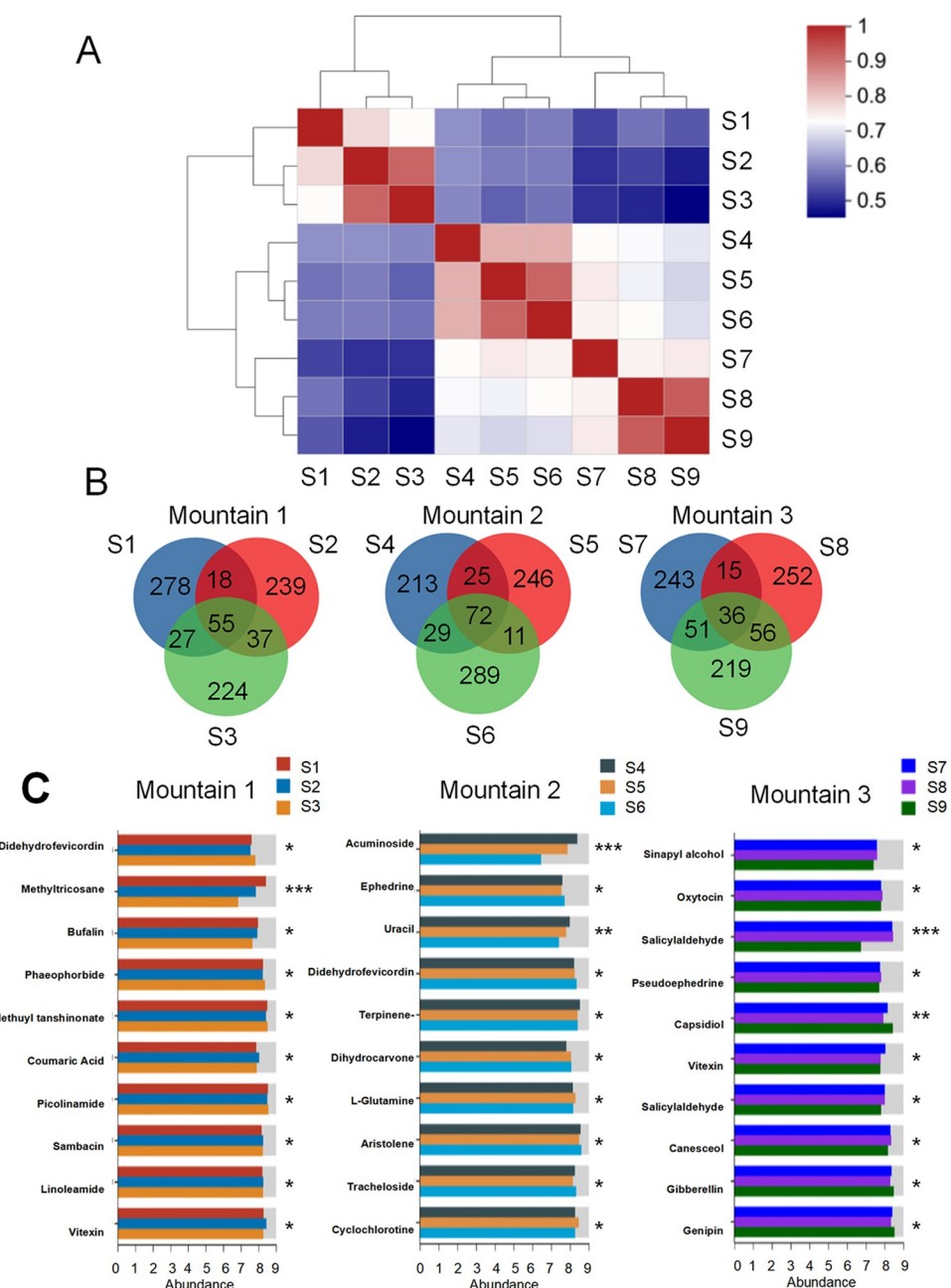

**Fig 3.** The right side and the lower side of the figure are sample names, each grid in the figure represents the correlation between two samples, different colors represent the relative size of the correlation coefficient between samples, the length of the clustering branch represents the relative distance between samples, and the samples on the same branch are similar (A). Detection of *Ephedra* metabolites in 9 *Ephedra* communities in three mountains (B). The content of the top ten differential metabolites was enriched (C).

shown in Fig 3(C), *E. saxatilis* on the three mountains showed differential metabolism, and the top ten metabolites of *E. saxatilis* on the three mountains were all inconsistent. The above results indicate that there are not only metabolic differences in the three mountains at different altitudes, but also significant differences in the three sampling points on the same mountain.

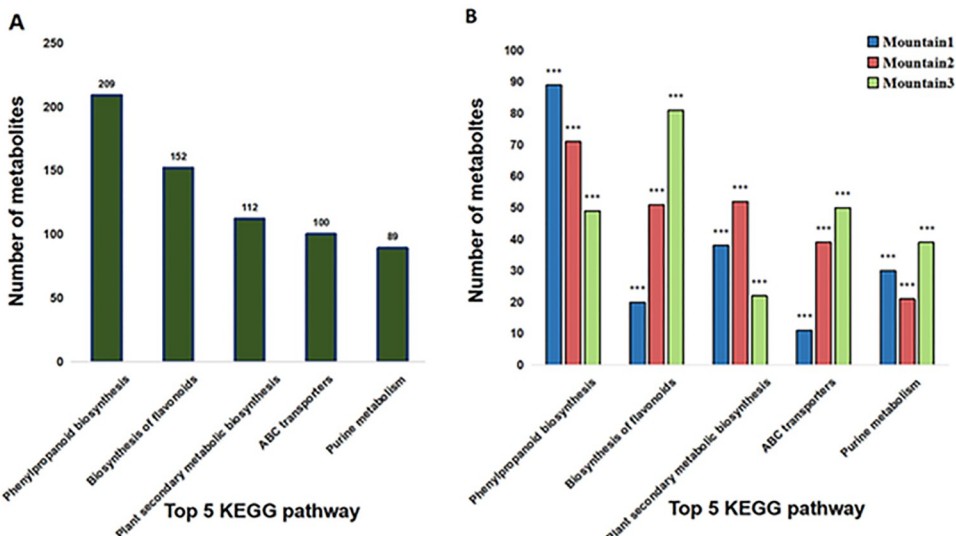

**Fig 4.** The total number of metabolites in the first five metabolic pathways (A). The top 5 KEGG enrichment pathways and number of metabolites in three mountains (B). Note: This quantity only represents the detection limit quantity.

## 3.4 The top 5 pathways of metabolite enrichment were analyzed in *Ephedra saxatilis*

There are many metabolic pathways in *Ephedra saxatilis*, so we only analyzed the top 5 pathways that are rich in metabolites, as shown in Fig 4(A). The first pathway is phenylpropanoid biosynthesis, which contains 209 metabolites. The second metabolic pathway was biosynthesis of flavonoids with 152 metabolites. The third pathway was plant secondary metabolic biosynthesis pathway, which contained 112 metabolites. The fourth largest is the ABC transports pathway, which contains 100 metabolites. In fifth place is the purine metabolism pathway with 89 metabolites. Of the five metabolic pathways in all communities, different communities have different amounts of metabolites. As shown in Fig 4(B), Mountain 1 has the most cluster metabolites in phenylpropanoid biosynthesis, it contains 89 metabolites. Mountain 3 has the most cluster metabolites in biosynthesis of flavonoids, it contains 81 metabolites. Mountain 2 has the most cluster metabolites in plant secondary metabolic biosynthesis, it contains 52 metabolites. Mountain 3 has the most cluster metabolites in ABC transports, it contains 52 metabolites. And Mountain 2 has the most cluster metabolites in purine metabolism, it contains 39 metabolites. There were significant differences in the metabolic quantities of the five metabolic pathways among different *Ephedra* communities. Therefore, there were differences in the diversity of *Ephedra* communities at different vertical altitudes.

## 3.5 Cluster analysis of metabolites in *Ephedra saxatilis*

As shown in Fig 5(A), the enrichment information of the first five metabolic pathways was screened out through cluster analysis of 9 samples from the three mountains. Each column in the figure represents a sample, and each row represents a metabolite. The color in the figure represents the relative expression amount of metabolites in this group of samples. For the specific change trend of the expression amount, please see the numbers under the color bar on the top left. Red represents a high concentration of metabolites and blue represents a low concentration of information. As shown in Fig 5(B), the pathway enrichment analysis of the first five pathways in nine communities on three mountains showed that the phenylpropanoid

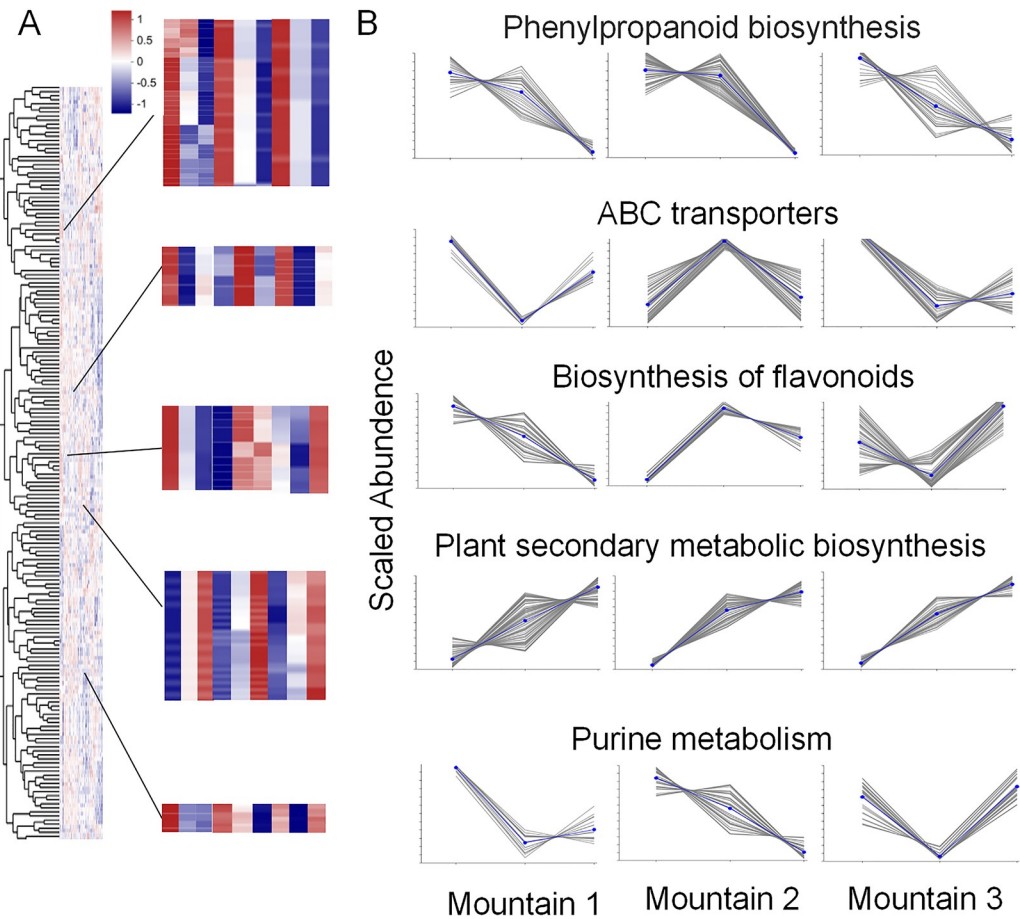

**Fig 5. Cluster analysis and elevation dynamic analysis of different metabolites of S1-S9 *Ephedra* from three mountains.** (A) The overall hierarchical clustering diagram in the figure normalized the enrichment values and clustered them, with red representing high metabolic enrichment and blue representing low metabolic enrichment; Color from red to blue, enrichment degree from large to small. (B) The figure represents the kinetic analysis of metabolites. The gray lines in each subgraph represent the relative enrichment of a metabolite in the community from different collection sites. The blue line shows the average enrichment of all metabolites in the community under different experimental conditions.

biosynthesis metabolic pathway showed an increasing trend with the increase of altitude sampling site. the plant secondary metabolic biosynthesis metabolic pathway all showed a downward trend with the increase of the sampled altitude. ABC pathway, Flavonoids pathway and Purine pathway show irregular changes with the altitude. The results show that not all pathways show regular changes.

## 3.6 Concentration of ephedrine and ephedrine in phenylpropanoid biosynthesis pathway

The next step is to analyze the metabolites of phenylpropanoid biosynthesis pathway. Eight annotated functional compounds were found to have high concentrations. As shown in Fig 6 (A), phenylpropanoid biosynthesis pathway analysis was performed on the samples from the three mountains, and only eight metabolites with high content were selected from each mountain, among which ephedrine and pseudoephedrine, catechol, and ferulic acid were highly enriched in the three mountains. As shown in Fig 6(B), ephedrine and pseudoephedrine

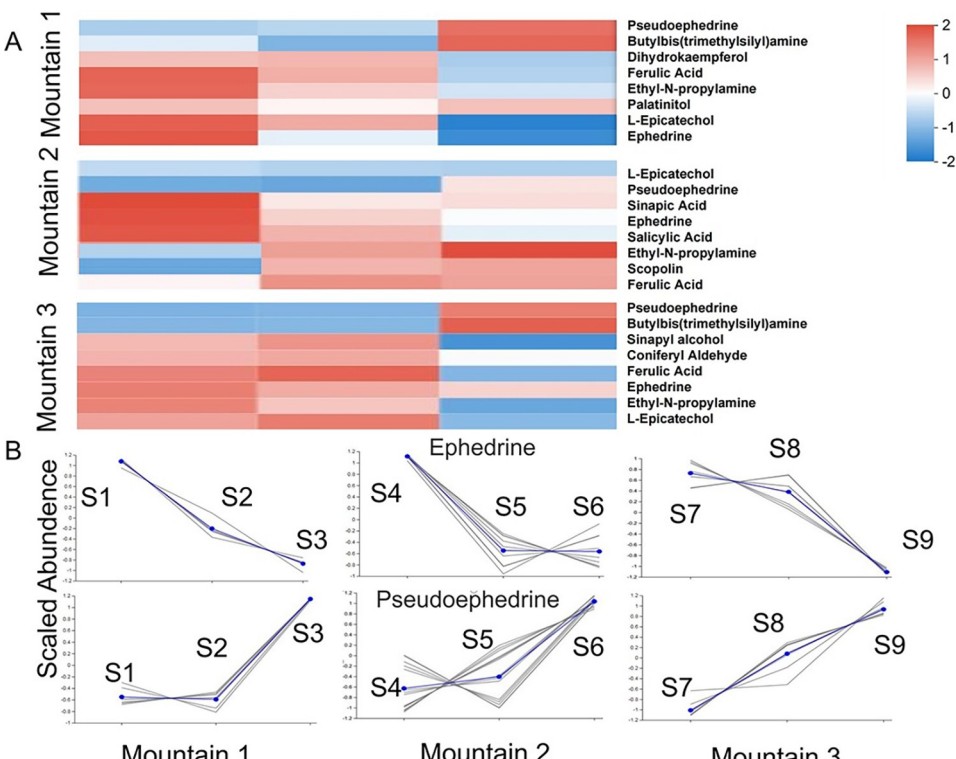

**Fig 6. Cluster analysis and dynamic analysis of metabolites in phenylpropanoid biosynthesis pathway in three mountains.** In Fig (A), the overall hierarchical clustering diagram normalized the enrichment values, with red representing high metabolic enrichment and blue representing low metabolic enrichment. The color ranges from red to blue and the enrichment degree from large to small. The bar length represents the contribution value of the metabolite to the difference between the groups, and the default value is no less than 5. The gray lines in Fig (B) indicate the relative enrichment of metabolites in communities from different collection points. The blue line is the average concentration of all metabolites in the community under different experimental conditions.

showed opposite trends in the three mountains, ephedrine content decreased with the increase of altitude, pseudoephedrine content increased with the increase of altitude gradient. Ephedrine content showed the lowest enrichment content at the highest elevations, such as S3,S6 and S9 in Fig 6(B). Pseudoephedrine exhibits the highest jealousy content at the highest elevations, as shown in Fig 6(B) (S1,S4,S7).

## 4 Discussion

The spatial differentiation of the Tibetan Plateau ecosystem is mainly determined by topographic features and atmospheric circulation patterns. Temperature and precipitation decrease to the northwest, resulting in a warm and humid climate in the southeast and a cold and dry climate in the northwest [7]. Therefore, plant growth environment has a variety of different landforms, in different landforms there are different sunlight exposure, soil humidity, temperature and other environmental factors, which will affect plant metabolism [12]. The main vegetation types in the southern study area of the Qinghai-Tibet Plateau are alpine meadow/scrub and alpine grassland in the alpine dry and cold area. However, even under the same climatic conditions, microenvironmental factors in alpine arid areas can lead to heterogeneity of vegetation. Under the influence of different climate factors, such as altitude, sunlight, longitude, latitude, humidity, temperature, etc., plants will also undergo some metabolic changes [25]. There are many kinds of *Ephedra* in Tibet. Many *Ephedra* varieties are not accurately identified

by appearance. The method of identifying the *Ephedra* by metabolism is to link the living environment and metabolism of *Ephedra* [26, 27]. In this study, it is found that altitude difference has a significant impact on soil stability, soil moisture and soil physical and chemical properties, so the landscape at different altitudes of the same mountain has unique vegetation. In this study, S3 was located on the sandy slope with the highest elevation, and the surface soil was prone to landslide and could not accumulate, and the water holding capacity of S3 was the lowest among the three communities (Table 2). On the other hand, there is a correlation between vegetation and water gradient in the study area. In the alpine arid region, soil water has a significant effect on plant growth and development, and the heterogeneity of soil water resources has a significant effect on plant community distribution and biomass [21]. Minami et al [6, 7] found that the content of ephedrine and pseudoephedrine would change somewhat with soil nutrients. However, for the change of environmental factors such as humidity and temperature, it is still unknown how ephedrine and pseudoephedrine changes with the change of environment. This study clarified that the contents of ephedrine and pseudoephedrine change with the change of altitude gradient. In order to further find out the relationship between Tibetan *Ephedra* and the environment, it is necessary to plant *Ephedra* under conditions of controlled soil moisture and temperature, so as to facilitate the composition ratio of ephedrine alkaloids in follow-up research. The collection sites of this study were mainly distributed in southern Tibet, and other areas may differ in plant distribution, environment and plant age.

Metabolomics is a rapidly developed interdisciplinary subject following genomics, transcriptomics and proteomics. In recent years, it has attracted wide attention and made important progress in the field of plant research, especially in the field of abiotic stress of medicinal plants, which is aimed at secondary metabolites [18, 19, 28, 29]. KEGG (Kyoto Encyclopedia of Genes and Genomes) is an online database where genomes, biochemical products, molecular interaction networks and other information are collected jointly [30–32]. In this study, KEGG was used to analyze the enrichment of five iconic metabolic pathways. The enrichment analysis of the top 5 pathways from 9 communities in 3 mountains showed that the phenylpropanoid biosynthesis pathway increased with the increase of the altitude of the sampling points. With the increase of sampling altitude, the secondary metabolic pathways showed a downward trend. The ABC pathway, flavonoid pathway and purine pathway showed irregular changes with altitude. It turns out that not all pathways change regularly. which also indicated that the increase of vertical altitude gradient caused the diversity and differentiation of metabolites in different communities. Plant metabolites can be roughly divided into primary metabolites and secondary metabolites. ABC transports metabolites support plant life secondary metabolites are essential for motility and growth, but they are more involved in environmental responses such as resistance to disease and stress [29, 33–35]. Secondary metabolism evolved from primary metabolism and plays an important role in many aspects of plant life. According to their structural properties, plant secondary metabolites can be divided into terpenoids, alkaloids, phenylpropanes and so on. In this study, with the increase of altitude, the metabolites of ABC transport pathway gradually increase, which also indicates that ABC transport pathway is adapting to the environment and changing itself with the adverse changes of the environment. With the increase of altitude, the metabolites in plant metabolic pathway first increased and then decreased, which also indicated that plant secondary metabolism may have the most adaptive conditions in the middle altitude. Purine metabolism pathway is the basic pathway to maintain the metabolism of the organism. The differentiation of the variation of the organism can be reflected by the degree of purine metabolism. Many studies on purine have also confirmed that the instability of purine metabolism can identify lesions and improve human health [36, 37]. We found that with the increase of vertical elevation gradient, the secondary metabolites in Ephedra community first decreased and then increased (Figs 4B and 5B), which

also indicated that the degree of variation was small in the middle of the altitude gradient, and the degree of differentiation was large at both ends. In this study, among the main secondary metabolite pathways in the vertical elevation ephedra community, the metabolic pathway of amphetamine was the first, and that of flavonoids was the second (Fig 4A). Ephedrine is a class of amphetamine compounds, including These include: L-ephedrine and D-pseudoephedrine, in addition to L-methylephedrine and D-methylpseudoephedrine, L-norephedrine, norpseudoephedrine, etc. which is the main active ingredient in *Ephedra* [38]. This also shows that *Ephedra* contains a lot of alkaloids. In this study, the phenylpropanoid biosynthesis pathway had the most metabolites enriched, and 209 metabolites were detected in 9 communities, indicating that this pathway plays an important role in the growth and development of *Ephedra*. This pathway contains ephedrine and pseudoephedra, as well as many other phenyl-C substances, indicating that there are still many metabolites of *E. saxatilis* that need to be functional verified. There are many compounds of medicinal value in flavonoids. These compounds are used to prevent and cure cardiovascular and cerebrovascular diseases, such as reducing the brittleness of blood vessels, improving the permeability of blood vessels, reducing blood lipid and cholesterol and so on [39–41]. In this study, the number of flavonoid metabolic pathways ranked second in *E. saxatilis* metabolism. The results showed that there were 152 flavonoid metabolites in *E. saxatilis*, and the proportion of flavonoid metabolites whose structure was identified and the efficacy and function were verified was relatively small, and the function and structure of many flavonoid metabolites still needed further development and utilization by scholars.

*Ephedra* has made many remarkable achievements in the clinical application of traditional Chinese medicine for thousands of years, and is listed as a medium product in Shennong Materia Medica [37]. Due to its strong sweating force and easy damage of healthy Qi, traditional Chinese medicine often regards it as a "tiger and Wolf". Proper use can play a therapeutic role, while improper use can produce toxic side effects. According to statistics, the application of *Ephedra* in contemporary Chinese medicine has a wide range, involving more than 50 diseases, especially the application of diseases in the respiratory system, circulatory system, head, facial features and limb joints [42]. In particular, ephedrine and pseudoephedrine have remarkable functions. According to the Chinese Pharmacopoeia (2020 edition) [43], the daily dose of *Ephedra* is 2 ~ 10 g. The total amount of ephedrine and pseudoephedrine contained in ephedrine should not be less than 0.8%. Different kinds of *Ephedra* have different total base content. Hong [44] identified five species of *Ephedra* in China and found that the total bases of three fluctuated significantly, and some were not included in the pharmacopoeia. But there is no specific report on the alkali of *E. saxatilis*. Therefore, our team will improve the content of alkali in *E. saxatilis* in the future. To provide a theoretical basis for the development and utilization of Tibetan medicine. Most of the ingredients in *E. saxatilis* are constantly being identified, but the quality of *Ephedra* is constantly changing in different environmental factors [7]. The individual base will gradually change with the change of metabolism. This is somewhat similar to Minami's [7] study of changes in ephedrine content in *Ephedra Gerardiana*. This may be because *Ephedra* has a similar altitude and climate on the plateau, or because the soil composition is similar. In this study, metabolomics method was used to analyze the metabolites of *E. saxatilis*, and the changes of ephedrine and pseudoephedrine with environmental factors were explained through the continuous classification of metabolites. The results showed that high metabolic communities could produce more ephedrine. The content of pseudoephedrine decreased with the increase of metabolite content. This may indicate that the total alkali content in *E. saxatilis* is relatively stable, maintaining the ecological stability of *E. saxatilis*. However, the sampling points of *ephedra* in this study are too few, and only limited conclusions can be drawn from a small range of *E. saxatilis*. In order to draw specific

conclusions on the changes of metabolic speed and ephedrine and pseudoephedrine content on the plateau, the sampling sites should be further expanded.

## 5 Conclusion

*E. saxatilis* was identified from a mountain in Upper Yadong County on the Tibetan Plateau and samples were collected at vertical elevation to analyze the metabolites in the growing soil and in vivo. By comparison with altitude and soil, we found that with the increase of altitude, soil moisture decreased and ephedra metabolites decreased significantly. Combined with metabolomic cluster analysis and PLS-DA and KEGG functional analysis, we can conclude that the increase of vertical elevation and the decrease of soil moisture lead to the slow down of *E. saxatilis* metabolism. This was confirmed by univariate analysis of the first five metabolites. Then, the first metabolic pathway was analyzed, and it was found that the content of ephedrine and pseudoephedrine changed with the metabolic speed of *E. saxatilis*. This indicates that the altitude can change the living environment of *E. saxatilis*, and the living environment will affect the speed of *E. saxatilis* metabolism, resulting in changes in the content of ephedrine and pseudoephedrine. The results of this study provide valuable information for further exploring the molecular mechanism of *E. saxatilis* metabolism and variation under special ecological environment, and also provide a useful reference for studying the developmental characteristics of *E. saxatilis*. At the same time, this study also provides a theoretical basis for how to improve the quality of *E. saxatilis* by adjusting the metabolism velocity.

## Supporting information

**S1 Dataset.**
(XLS)

## Author Contributions

**Conceptualization:** Mengnan Lu, Shiyan Wang, Xiaona Wang.

**Data curation:** Mengnan Lu, Xiaona Wang.

**Formal analysis:** Mengnan Lu.

**Investigation:** Mengnan Lu.

**Methodology:** Mengnan Lu.

**Project administration:** Mengnan Lu.

**Resources:** Mengnan Lu, Shiyan Wang.

**Software:** Mengnan Lu.

**Supervision:** Mengnan Lu.

**Validation:** Mengnan Lu, Yongjuan Zhang, Shengnan Zhang.

**Writing – original draft:** Ji De.

**Writing – review & editing:** Ji De.

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
