## [Decision Letter · Decision Letter 0]

12 Jun 2023

PONE-D-23-11368Ephedrine and pseudoephedrine in Ephedra saxatilis on the vertical altitude gradient changed in southern Tibet Plateau, ChinaPLOS ONE

Dear Dr. Mengnan Lu,

Thank you for submitting your manuscript to PLOS ONE. After careful consideration, we feel that it has merit but does not fully meet PLOS ONE’s publication criteria as it currently stands. Therefore, we invite you to submit a revised version of the manuscript that addresses the points raised during the review process.

We look forward to receiving your revised manuscript.

Kind regards,

Alia Ahmed

Academic Editor

PLOS ONE

Journal Requirements:

"YES - Specify the role(s) played.Our work was partly financially supported by the National Natural Science Foundation of China（32060087."

"YES - Specify the role(s) played.Our work was partly financially supported by the National Natural Science Foundation of China（32060087."

7. We note that Figure 1 in your submission contain copyrighted image. All PLOS content is published under the Creative Commons Attribution License (CC BY 4.0), which means that the manuscript, images, and Supporting Information files will be freely available online, and any third party is permitted to access, download, copy, distribute, and use these materials in any way, even commercially, with proper attribution. For more information, see our copyright guidelines: http://journals.plos.org/plosone/s/licenses-and-copyright.

Reviewers' comments:

Reviewer's Responses to Questions

**Comments to the Author**

1. Is the manuscript technically sound, and do the data support the conclusions?

Reviewer #1: Yes

Reviewer #2: Partly

2. Has the statistical analysis been performed appropriately and rigorously? 

Reviewer #1: N/A

Reviewer #2: Yes

3. Have the authors made all data underlying the findings in their manuscript fully available?

Reviewer #1: Yes

Reviewer #2: Yes

4. Is the manuscript presented in an intelligible fashion and written in standard English?

Reviewer #1: Yes

Reviewer #2: Yes

5. Review Comments to the Author

Reviewer #1: 1. It has not been explained in the text the basic justification for the difference in height used in this experiment. the reason why it was chosen from these 3 mountains might be mentioned.

2. The synthesis of metabolite compounds in plants can be influenced by many factors such as light intensity, soil nutrition, temperature and etc. as well as other factors in the process of plant growth. In this study, there is no justification regarding the influence of other parameters besides height difference in this study. Explanation of other factors that may affect the synthesis of secondary metabolites in Ephedra s. can be added in paper.

3. The 9 samples were taken, 3 samples each from different mountains. It is better if the notation used for each sample is consistent. In figure 3 (B) S1-S9 is used, while in figure 6 (B) S1-S3 for mountain 1, mountain 2 and 3. This can confuse the reader in understanding the findings

4. Data processing and analysis needs to be explained in more detail. In accordance with the data displayed in the results section. Especially the method used to analyze the LC/MS-MS results so that it can be found a pattern of changes in the concentration of ephedrine and pseudoephedrine which changes according to the difference in the height of the mountains where the samples were taken.

Reviewer #2: This manuscript examines Ephedrine and pseudoephedrine in Ephedra saxatilis in the southern Tibet Plateau, China which is still rarely done. The introductory section should be arranged systematically, explaining why the importance of this research is carried out. The method section should be explained in more detail regarding the selected method. In the discussion section, it is better to explain more comprehensively the results obtained.

6. PLOS authors have the option to publish the peer review history of their article (what does this mean?). If published, this will include your full peer review and any attached files.

Reviewer #1: **Yes: **Elvira Yunita

Reviewer #2: **Yes: **Rauza Sukma Rita

---

## [Author Response · Author response to Decision Letter 0]

13 Jul 2023

Dear Revewer:

We submit our manuscript entitled “Ephedrine and pseudoephedrine in Ephedra saxatilis on the vertical altitude gradient changed in southern Tibet Plateau, China” for publication in PLoS ONE. Thank you for receiving your valuable comments, We agree with them very much and the following modifications have been made:

1.Sample collection

The reason why we chose the three mountains is that the vertical altitude gradient of the three mountains is within 500 meters, and the highest altitudes of the three mountains are different. In order to study the impact of altitude changes on ephedrine and pseudoephedrine, when the altitude range is similar, the reliability and stability of the data can be guaranteed. In the sample collection part, we added the reasons why we chose three mountains. Although the more sample mountains is better, it is very difficult to find similar mountains in high-altitude areas. In the next test, we will expand the collection site and find more sampling points.

2.Effects of environmental factors on Ephedra saxatilis

In our study, we found that altitude has an impact on ephedrine and pseudoephedrine, and environmental factors may also be factors for the difference in E. saxatilis metabolism. In the discussion section, it was proposed that soil moisture, light and temperature may affect the metabolism of E. saxatilis. Since there are only three mountains sampled in this study, and the environmental factors on each mountain are similar, the analysis of environmental factors and E. saxatilis in small samples may lead to inaccurate results. At present, we are expanding the amount of E. saxatilis collection in a wider range of altitude gradiants to collect environmental factors such as soil moisture, temperature and humidity, hoping to find out the influence of environmental factors on E. saxatilis in further studies.

3.Figure 6 modification

Thank you for your comments on Figure 6B, we have revised Figure 6B under your guidance. In figure 6 (B) S1-S9 for mountain 1, mountain 2 and 3.

4.Data processing and analysis

We have made a more detailed explanation of the data processing, including the application of software, the analysis and comparison of databases, and provided the corresponding website. We analyzed the results in more detail, including the amount of metabolites and changes with altitude, as well as ephedrine and pseudoephedrine with altitude. A large number of analysis results were added in the discussion section, indicating that phenylpropanoid biosynthesis pathway is the most important pathway in E. saxatilis, and there are still many chemicals in E. saxatilis that have not been identified in structure and pharmacological effects.

7.Introduction section

We have made a systematic arrangement in the introduction part again, emphasizing the importance of this study. Although there are many effective chemicals in Ephedra, ephedrine and pseudoephedrine are the most important medicinal ingredients in Ephedra. At the same time, the use of metabolomics to analyze Ephedra metabolism and find important metabolites is emphasized, which provides convenience for ecology and plateau plant research.

8.Method section

We describe the method in more detail, including the preparation process of the preliminary sample and the sample injection process. 

9.Discussion section

 In the discussion section, we analyze all the results more. Combining the data and changes in the natural environment, we analyzed a number of possibilities and found that altitude has an impact on the content of ephedrine and pseudoephedrine. However, we still do not know whether some environmental factors, such as soil moisture and air temperature, will affect the content of ephedrine and pseudoephedrine. Therefore, in the next study, we will expand the sampling sites and analyze the influence of climate factors on E. saxatilis.

---

## [Decision Letter · Decision Letter 1]

15 Aug 2023

Ephedrine and pseudoephedrine in Ephedra saxatilis on the vertical altitude gradient changed in southern Tibet Plateau, China

PONE-D-23-11368R1

Dear Dr. Lu,

We’re pleased to inform you that your manuscript has been judged scientifically suitable for publication and will be formally accepted for publication once it meets all outstanding technical requirements.

Kind regards,

Alia Ahmed

Academic Editor

PLOS ONE

Additional Editor Comments (optional):

Reviewers' comments:

Reviewer's Responses to Questions

**Comments to the Author**

1. If the authors have adequately addressed your comments raised in a previous round of review and you feel that this manuscript is now acceptable for publication, you may indicate that here to bypass the “Comments to the Author” section, enter your conflict of interest statement in the “Confidential to Editor” section, and submit your "Accept" recommendation.

Reviewer #1: (No Response)

Reviewer #2: All comments have been addressed

2. Is the manuscript technically sound, and do the data support the conclusions?

Reviewer #1: (No Response)

Reviewer #2: Yes

3. Has the statistical analysis been performed appropriately and rigorously? 

Reviewer #1: (No Response)

Reviewer #2: Yes

4. Have the authors made all data underlying the findings in their manuscript fully available?

Reviewer #1: (No Response)

Reviewer #2: Yes

5. Is the manuscript presented in an intelligible fashion and written in standard English?

Reviewer #1: (No Response)

Reviewer #2: Yes

6. Review Comments to the Author

Reviewer #1: (No Response)

Reviewer #2: (No Response)

7. PLOS authors have the option to publish the peer review history of their article (what does this mean?). If published, this will include your full peer review and any attached files.

Reviewer #1: **Yes: **Elvira Yunita

Reviewer #2: **Yes: **dr.Rauza Sukma Rita, Ph.D

---

## [Editor Report · Acceptance letter]

18 Aug 2023

PONE-D-23-11368R1 

Ephedrine and pseudoephedrine in *Ephedra saxatilis* on the vertical altitude gradient changed in southern Tibet Plateau, China 

Dear Dr. De:

I'm pleased to inform you that your manuscript has been deemed suitable for publication in PLOS ONE. Congratulations! Your manuscript is now with our production department. 

Kind regards, 

on behalf of

Dr. Alia Ahmed 

Academic Editor

PLOS ONE